# New Kids on the Block: Bile Salt Conjugates of Microbial Origin

**DOI:** 10.3390/metabo12020176

**Published:** 2022-02-13

**Authors:** Ümran Ay, Martin Leníček, Arno Classen, Steven W. M. Olde Damink, Carsten Bolm, Frank G. Schaap

**Affiliations:** 1Department of General, Visceral and Transplant Surgery, University Hospital Aachen, 52074 Aachen, Germany; uay@ukaachen.de (Ü.A.); steven.oldedamink@maastrichtuniversity.nl (S.W.M.O.D.); 2Institute of Medical Biochemistry and Laboratory Diagnostics, Faculty General Hospital and 1st Faculty of Medicine, Charles University, 12808 Prague, Czech Republic; mleni@centrum.cz; 3Institute of Organic Chemistry, RWTH Aachen University, 52074 Aachen, Germany; arno.classen@oc.rwth-aachen.de (A.C.); Carsten.bolm@rwth-aachen.de (C.B.); 4Department of Surgery, NUTRIM School of Nutrition and Translational Research in Metabolism, Maastricht University, 6200 MD Maastricht, The Netherlands

**Keywords:** bile salts, microbial metabolites, bile salt receptor, bile salt signaling, gut microbiota

## Abstract

Biotransformation of host bile salts by gut microbes results in generation of secondary bile salt species that have biological and physicochemical properties that are distinct from the parent compounds. There is increased awareness that a bile salt–gut microbiome axis modulates various processes in the host, including innate and adaptive immunity, by interaction of microbial bile salt metabolites with host receptors. Omics and targeted approaches have vastly expanded the number and repertoire of secondary bile salt species. A new class of microbial bile salt metabolites was reported in 2020 and comprises bile salts that are conjugated by microbial enzymes. Amino acids other than those employed by host enzymes (glycine and taurine) are used as substrates in the formation of these microbial bile salt conjugates (MBSCs). Leucocholic acid, phenylalanocholic acid and tyrosocholic acid were the first MBSCs identified in mice and humans. The number of distinct MBSCs is now approaching 50, with variation both at the level of bile salt and amino acid employed for conjugation. Evidence is emerging that MBSC generation is a common feature of human gut bacteria, and initial links with disease states have been reported. In this review, we discuss this intriguing new class of secondary bile salts, with yet enigmatic function.

## 1. Introduction

The gut is a complex ecosystem that harbors a wide variety of microorganisms that are intricately linked to host health and disease. To communicate with the host, the microbiome converts diet- and host-derived molecules into metabolites with signaling function. These signaling molecules include bile salts, amphipathic molecules produced in the liver that are the active ingredient of the digestive fluid bile [1]. Bile salts are biological detergents that promote digestion, solubilization and absorption of dietary lipids, including lipid-soluble vitamins, in the small intestine. Bile salts undergo active reabsorption in the distal part of the small intestine for recirculation to the liver. The small fraction of bile salts that escapes reabsorption in the ileum enters the colon, where microbial metabolism of bile salts takes place. These so-called secondary bile salts can be passively absorbed by the colonic epithelium and are carried to the liver by the portal circulation. The cyclic process of release of bile salts in bile, flow through the biliary network and intestinal lumen and portal venous return to the liver is termed the enterohepatic circulation. Dedicated membrane transporters at sites of transcellular passage are key to the efficiency of this process [1]. Apart from their digestive function, bile salts have antimicrobial activity and are sensed by dedicated host receptors that control a variety of metabolic and biological processes. The interaction between gut microbes and bile salts thus impacts host (patho) physiology. In this review, we discuss a newly discovered class of microbial bile salt metabolites with yet enigmatic function, *viz*. microbial bile salt conjugates, and present preliminary experimental data on this intriguing new class of bile salts.

### 1.1. Hepatic Bile Salt Conjugation

In humans, the liver produces two major bile salts from the membrane lipid cholesterol, namely cholic acid (CA) and chenodeoxycholic acid (CDCA), while the rodent liver can additionally convert CDCA into muricholic acids (αMCA and βMCA). These host-derived bile salts are termed primary bile salts, and the biosynthetic pathways have been eloquently reviewed elsewhere [2]. Their general molecular structure comprises a steroid nucleus of four fused rings containing one or multiple hydroxyl substitutions and a hydrocarbon side chain with a terminal carboxylic acid group (Figure 1A). Before their release in bile, this carboxylic acid moiety is conjugated (i.e., *N*-amidated) with the amino acid glycine or taurine [3]. Hence, bile salts entering the biliary network and eventually the proximal small intestine are principally present as conjugated species. The lowering of the pKa value following *N*-amidation of the side chain causes bile salts to be predominantly present in ionized form at the ambient pH in their enterohepatic trajectory [1]. This permanent negative charge, as well as size, prevents passive membrane permeation of bile salts in the proximal small intestine where bile salt transporters are not expressed [4]. Consequently, a high luminal concentration is maintained in which bile salts are present in a micellar phase that is essential for its digestive functions. Along with relatively rapid gastrointestinal transit and local environmental factors (e.g., oxygen and pH), the bactericidal action of bile salts (present in millimolar concentrations) may contribute to low numbers of bacteria in this region of the intestine [1].

In the human liver, conjugation of bile acids involves two enzymatic reactions. Bile acids are first converted to a reactive thioester intermediate by bile acid CoA synthetase (BACS). Next, bile acid CoA: amino acid *N*-acyltransferase (BAAT) catalyzes the transfer of the bile acid moiety to glycine or taurine, thus forming an amide bond. Selection of taurine and glycine as exclusive amino acid substrates in hepatic *N*-amidation of bile salts relates to the resistance of the respective conjugates to degradation by host enzymes encountered in their enterohepatic trajectory, e.g., carboxypeptidases in pancreatic juice and membrane-bound (carboxy) peptidases in the small intestinal mucosa [5]. This concept arose from studies with synthetic bile salt conjugates that were susceptible to cleavage by pancreatic carboxypeptidase unless glycine or taurine was used for conjugation, with conjugates of these particular amino acids being largely inert to enzymatic degradation [5]. As detailed below, glycine- and taurine-conjugated bile salts can be hydrolyzed (i.e., deconjugated) by microbial bile salt hydrolases (BSHs), which in quantitative terms is an important pathway in the microbe-dense colon in particular [6].

Bile salt *N*-amidation affects the acidic strength of the compound, i.e., the propensity to dissociate into a bile salt anion and a proton, increasing the amphipathic character of the bile salt. This effect is most pronounced for conjugation with taurine, with pKa values typically three pH units lower than the unconjugated parent molecule. As a consequence of conjugation, bile salts are thus largely present as negatively charged molecules at ambient pH, improving their aqueous solubility and rendering them largely membrane-impermeant in the absence of a transport protein [7]. Active and passive cellular uptake of (un)conjugated bile salts is discussed in greater detail in Section 2.1.

### 1.2. Microbial Bile Salt Conjugation

Colonic spill-over brings bile salts in direct contact with a microbe-dense environment, resulting in their biotransformation. Apart from removal of the amino acid moiety of conjugated bile salts, three additional reaction types are known to be catalyzed by microbial enzymes and lead to formation of so-called secondary bile salts. These reactions include removal (dehydroxylation), oxidation (dehydrogenation) and epimerization of nuclear hydroxyl groups [6]. Deconjugation of bile salts is a prerequisite for 7α-dehydroxylation of cholic acid and chenodeoxycholic acid to form deoxycholic acid (DCA) and lithocholic acid (LCA), respectively, and is catalyzed by bacterial BSHs. Excellent reviews on microbial bile salt metabolism and the enzymes involved can be found elsewhere [6,8].

A fifth reaction type was reported by Quinn et al. in 2020 [9]. Through an elegant mass spectrometric approach aimed at identifying bacterial metabolites on a whole-body scale in mice, atypical conjugates of CA with leucine, phenylalanine and tyrosine were identified and shown to be of microbial origin. These MBSCs were readily detected in the intestines, particularly in the small intestine, of specific pathogen-free but not germ-free mice. Initial screening of cultures of human gut bacteria revealed that two *Clostridium bolteae* (now known as *Enterocloster bolteae*) strains were able to produce PheCA and TyrCA when presented with unconjugated CA and the respective amino acid precursor [9]. A recent study by Lucas et al. uncovered a much larger diversity of MBSCs in terms of bile acid and amino acid precursors, with both primary (CA, CDCA) and secondary (DCA, LCA) bile salts used for conjugation with one of 15 proteinogenic amino acids in bacterial cultures [10]. Among 44 newly identified MBSCs, conjugates with phenylalanine and glycine were frequently observed. Furthermore, this study revealed that microbial (re) conjugation of bile salts appears to be a common feature, with 27 of 70 bacterial species common to the human gut capable of doing so. Bacterial producers included both Gram-negative and Gram-positive species [10].

It is conceivable that the physicochemical properties of the amino acid employed for conjugation affect the physicochemical and biological properties of the resultant bile salt conjugate. For instance, leucine and phenylalanine are hydrophobic amino acids, and the bile salt conjugates thereof may be more hydrophobic than the glycine equivalent formed by the host. The enzymes involved in microbial bile salt conjugation have not been identified yet. A two-step biosynthetic pathway akin to the process in mammalian liver is conceivable. Orthologs of mammalian BACS and/or BAAT were not apparent from in silico analyses of the genomes of *C. bolteae* strains. It is plausible that BSH activity yields the unconjugated bile acids that are the likely substrates in the biosynthetic route. Thus far, it has not been explored if bacterial producers rely on endogenous BSH activity or bacterial interplay for formation of MBSCs in vivo. Unconjugated bile acids may also originate from partial degradation of glycine-conjugated bile salts by pancreatic carboxypeptidases. Amino acid cosubstrates should be available from multiple sources, including digested dietary proteins and bacterial biosynthesis. Bacterial import of (de)conjugated bile salt precursors and release of de- and reconjugated bile salts is poorly characterized. Most likely, carrier proteins are required for the bidirectional movement of (un)conjugated bile salts across the outer and inner membrane of bacteria. Bacteria capable of 7α-dehydroxylation of bile salts such as *Clostridium scindens* express a membrane transporter (*baiG*) for energy-dependent uptake of preferably unconjugated primary bile acids [11]. Another putative bile salt transporter has been found in *Lactobacillus johnsonii* [8]. In Gram-negative bacteria, the general porin *ompF* is implicated in bile salt uptake across the outer membrane. Diffusion of uncharged protonated bile acids across the membrane(s) may contribute to bacterial uptake of bile salts [8].

Cloning of the proteins involved in MBSC formation (and release) will shed further light on substrate specificity, cofactors and bioenergetic requirements. It is fair to assume that ATP is directly consumed (and is indirectly not generated by degrading the amino acid used for conjugation) in the process, implying a reason why certain bacteria generate these compounds.

## 2. Possible Functions of MBSCs

At present, little information is available on the biological functions of MBSCs. They may have a function in the shaping of gut microbial composition. Analogous to their host counterparts, MBSCs may activate bile salt receptors in host tissues. Unless action is restricted to the gut lumen, the latter would require intestinal absorption and distribution further in the body. The transporters involved in the enterohepatic circulation of bile salts are briefly discussed first.

### 2.1. MBSCs Are Likely Substrates for Bile Salt Transporters

Dedicated transporters are the cornerstone of the enterohepatic circulation of bile salts. In the terminal part of the small intestine, bile salts are reclaimed in a sodium-dependent manner via the apical sodium-bile salt transporter (ASBT), which is encoded by the *SLC10A2* gene and is expressed at the enterocyte brush border membrane. Both *N*-amidated and unconjugated bile salts are transported by ASBT, with a clear preference for the former [12]. Mono-, di- and trihydroxy bile salt species are all substrates for ASBT and individually compete for binding to ASBT [13]. Substrate preference of ASBT, as well as regulation of its expression, may vary between animal species [14].

An early study on the intestinal bile salt uptake system defined structural requirements for transport of natural bile salts and synthetic analogs in everted gut sacs [15]. Bile salts with a single negative charge in the conjugate moiety were effectively transported, while a positive charge or a second negative charge prevented uptake. Hence, it is likely that MBSCs meeting these demands (e.g., LeuCA, PheCA and TyrCA) are substrates for ASBT. Apart from active transport, absorption of neutral (i.e., protonated) bile salts, both in their free and conjugated form, occurs along the entire length of the small intestine and colon [12]. Hence, passive permeation can be expected for MBSCs with weak acid properties that leave a fraction of the molecules in the uncharged state at the slightly acidic pH at the cell surface. In the setting of intestinal barrier dysfunction, disruption of intercellular junctions may allow portal entry of luminal molecules such as MBSCs through enhanced paracellular diffusion [16].

After entering the cytosolic compartment of the enterocyte, bile salts can bind to the ileal lipid-binding protein (ILBP) and be carried to the basolateral membrane for release into the portal circulation by the heterodimeric transporter OSTα/β. Transport via OSTα/β is sodium independent and presumably bidirectional. Transport direction may thus depend on the substrate gradient [17]. Apart from bile salts, OSTα/β can transport endogenous lipids, including sulfated steroids, neurosteroids and prostaglandins. This broader substrate specificity makes it realistic that OSTα/β also releases MBSCs in the portal circulation.

In the liver, extraction of conjugated bile salts from the sinusoidal blood is accomplished by Na^+^ taurocholate cotransporting protein (NTCP). NTCP is the first member of the SLC10A subfamily of transporters that includes ASBT, as well as five additional members with poorly charted, if any, transport characteristics. As its abbreviation implies, transport via NTCP is sodium dependent. Substrates also include molecules other than bile salts, e.g., thyroid hormones and drugs. In a perfused liver setup, sidechain charge was identified as a major determinant of uptake of natural and synthetic bile salt conjugates. Monovalent and divalent bile salt anions were effectively taken up, while cationic or zwitterionic variants showed negligible uptake [18]. It is thus conceivable that MBSCs that meet these requirements are substrates for NTCP. This notion is supported by empirical evidence (Section 2.2.2, Figure 2). Again, passive uptake of protonated MBSCs with weak acid properties can be expected to contribute to some extent to hepatic uptake. In the cytoplasm of the hepatocyte, carriers such as fatty acid-binding protein 1 (L-FABP) are involved in delivery to the apical membrane for secretion into the canalicular space. Biliary bile salt release is against a steep concentration gradient and driven by ATP-dependent transport via the bile salt export pump (BSEP), which has a preference for monovalent *N*-amidated bile salts [19]. Transport of synthetic bile salt conjugates by BSEP has not been studied, but it is likely that at least MBSCs with a single negative charge are BSEP substrates.

The respective transporter characteristics support the notion that MBSCs have the potential to be absorbed and undergo enterohepatic cycling, implying that MBSCs have the ability to interact with host bile salt receptors outside the gut lumen.

### 2.2. MBSCs Are Likely Ligands for Bile Salt Receptors

Host bile salt receptors can be grouped in receptors that are expressed at the plasma membrane and in intracellular receptors that rely on cellular ligand entry for their function.

#### 2.2.1. Plasma Membrane Bile Salt Receptors

At least three distinct G-protein-coupled receptors have been shown to be activated by bile salts, with ligand specificity ranging from broad to confined to particular bile salt species. The first membrane receptor that was identified was TGR5/GPBAR1 [20]. TGR5 is widely expressed in tissues, with tissue expression typically restricted to certain cell types, including those exposed to bile salts as part of their enterohepatic (e.g., liver and intestine) or systemic circulation (e.g., skeletal muscle, brain) [21,22]. TGR5 is a dedicated bile salt receptor and has a greater affinity for secondary than for primary bile salts (LCA > DCA > CDCA > CA) and for conjugated over unconjugated species [23,24,25]. EC_50_ values for activation of TGR5 by endogenous bile salts, assayed by diverse methods, range from 0.03–3.7 µmol/L for LCA to 7.7–27 µmol/L for CA [24]. *N*-amidation, especially with taurine, results in higher receptor affinity, as showcased by TLCA, GLCA and LCA, with EC_50_ values of 0.29, 0.54 and 0.58 µmol/L, respectively [25].

Interestingly, preliminary experiments by our group revealed that MBSCs (i.e., Leu(CD)CA, Phe(CD)CA and Tyr(CD)CA) were able to activate TGR5, with affinities somewhat lower than their respective glycine and taurine conjugates (Ay, Haider, Hoffman, Schaap et al., unpublished observations). Ligand binding to TGR5 stimulates adenylate cyclase activity resulting in elevated cAMP levels and downstream signaling events [26]. TGR5 participates in the innate immune response by dampening LPS-induced cytokine release from macrophages. TGR5 has additional roles among other in glucose homeostasis and regulation of energy expenditure.

At least two other membrane receptors that are activated by high-affinity endogenous ligands also respond to bile salts. These comprise muscarinic receptor type 3 (M3R) and the sphingosine-1-phosphate receptor 2 (S1PR2). M3R is expressed among other in cells of the enteric nervous system and gastrointestinal epithelia [27]. This receptor is selectively activated by taurolithocholic and the sulfated form thereof, which act as partial agonists and have been proposed to structurally resemble the high-affinity ligand acetylcholine. EC_50_ values for binding of bile salt ligands to M3 have not been reported, but the minimal dose of taurolithocholic acid to elicit effects in cells overexpressing M3 ranges from 10 to 250 µmol/L [27]. Unconjugated LCA and other secondary or primary bile salts do not activate M3R. The only LCA-based MBSC reported so far is glycolithocholic acid (GLCA), which has unknown activity on M3R and can also be formed by the host. Ligand binding to M3R results in activation of phospholipase C and formation of the secondary messengers diacylglycerol and inositol-1,4,5-triphosphate. Insights into the consequences of M3R agonism by LCA conjugates primarily stem from in vitro studies in cultured cells and have been reviewed elsewhere [28]. It remains to be determined whether MBSCs can activate M3R, with bile salt selectivity of the receptor not arguing in favor of such action.

S1PR2 is one of the receptors for sphingosine 1-phosphate, a bioactive lipid implicated among other in cell proliferation and cell survival [29]. Conjugated, but not unconjugated, bile salts have been identified as agonists of S1PR2 (no reported EC_50_ values), with both primary and secondary bile salt conjugates triggering activation of downstream AKT and ERK_1/2_ MAPK signaling cascades in an S1PR2-dependent fashion in isolated hepatocytes. S1PR2 is ubiquitously expressed, but bile salt modulation of this receptor may be restricted to tissues exposed to high levels of conjugated bile salts such as the liver. Here, S1PR2 is highly expressed in hepatocytes and to a lesser extent in cholangiocytes, sinusoidal endothelial cells, myofibroblasts and macrophages. In the liver, S1PR2 is involved in regulation of lipid metabolism and inflammation and has been implicated in numerous liver diseases, including disorders characterized by bile salt accumulation such as cholestatic liver disease and malignancies of the bile ducts (for reviews, see [29,30]). Given its bile salt ligand specificity, it is conceivable that bile salt conjugates of microbial origin are also able to activate S1PR2.

#### 2.2.2. Nuclear Bile Salt Receptors

The farnesoid X receptor (FXR) is the prototypical intracellular receptor for bile salts and belongs to the superfamily of nuclear hormone receptors. These are ligand-activated transcription factors activated by a range of lipophilic compounds of endogenous and exogenous origin. FXR is a dedicated receptor for bile salts and is widely expressed, with particularly high expression in the small intestine and liver. Binding affinity is highest for hydrophobic bile salts (CDCA > DCA = LCA > CA) and is not affected by conjugation status [31]. CDCA is the most potent endogenous ligand with an EC_50_ value of 4.5–13 µmol/L [25,31]. Agonist binding to FXR results in a conformational change and displacement of bound corepressors and recruitment of transcriptional coactivators [32]. FXR target genes are numerous, and the various biological processes regulated by FXR have been the subject of excellent reviews [33].

A main function highlighted here is the central role of FXR in maintaining intracellular bile salt levels in a safe range and accordingly safeguarding against toxic consequences of bile salt accumulation [34]. This vital homeostatic role is exerted at multiple levels, including a gut–liver signaling axis that controls hepatic bile salt synthesis via a negative feedback loop. Thus, bile salts released in the jejunum after meal-induced gallbladder contraction repress their own synthesis by stimulating ileal transcription of FXR target FGF19/Fgf15. This endocrine factor reaches the liver via the portal circulation, where it activates a specific receptor complex (FGFR4/βKlotho) at the cell surface of hepatocytes. Downstream signaling cascades culminate in downregulation of the gene (i.e., *CYP7A1*) coding for the rate-limiting enzyme in the bile salt biosynthetic pathway [33].

In the initial report on MBSCs, activation of FXR by LeuCA, PheCA and TyrCA was studied in vitro and in vivo [9]. Using a cell reporter approach in HEK293 embryonic kidney cells, TyrCA and PheCA were identified as stronger agonists of FXR than CDCA, the most potent endogenous activator of FXR (EC_50_ values: 0.14 µM, 4.5 µM and 9.7 µM, respectively). LeuCA was unable to elicit FXR activation in this experimental setup, which relied on endogenous expression of transporters capable of importing the test compounds and/or passive permeation of MBSCs [9]. Using human HuH7 hepatoma cells overexpressing human NTCP, we observed induction of FXR target genes by LeuCA, PheCA and TyrCA (Figure 2) (Ay, Schaap et al., unpublished observations). In mock-transfected HuH7 cells, FXR target gene induction was only noted for obeticholic acid (data not shown), a semi-synthetic potent FXR agonist that is membrane permeable. This indicates that the tested MBSCs are substrates for NTCP, in line with the empirical substrate requirements discussed earlier.

Mice receiving LeuCA or TyrCA by daily gavage (500 mg/kg body weight) had enhanced intestinal expression of Fxr target genes *Fgf15* and *Shp* after 24 h (2 dosages), with a similar extent of upregulation observed in mice administered the parent bile salt (CA) [9]. This was accompanied by reduced expression of *Cyp7a1* in the liver of mice receiving MBSCs or CA, indicating repression of bile salt synthesis through the intestinal Fxr/Fgf15 axis. Hepatic Fxr target gene induction was observed after repeated gavage (4 dosages, 72 h), with elevated expression of *Shp* seen in mice receiving MBSCs [9]. These findings suggest that MBSCs are absorbed in the small intestine and engage with Fxr in the ileum and liver. It cannot be ruled out though that transcriptional effects of MBSCs were due to the parent bile salt (CA) after cleavage of the *N*-amidate bond in MBSCs by host and/or microbial hydrolases. In an unrelated animal experiment, MBSCs were readily detected by a sensitive LC-MS assay in intestinal content and fecal matter, while compounds were not detectable in the portal or systemic blood [9]. As detailed below, we found that MBSCs are indeed prone to degradation by enzymes present in the lumen of the intestine or its microbiota, and this may explain the apparent failure to detect intact MBSCs in the bloodstream. It is also conceivable that MSBCs undergo further microbial conversion and accordingly escape detection by LC-MS. Formation of oxidized forms (keto bile salts) of LeuCA, PheCA and TyrCA was noted upon incubation with fecal cultures [9]. Further investigations should shed light on intestinal absorption and a possible enterohepatic circulation of MBSCs and metabolites thereof.

Nuclear receptors besides FXR can be activated by bile salts. These alternative receptors generally bind only a limited number of bile salt species, typically metabolites of microbial origin. Among others, the transcriptional response evoked by these receptors can be broadly categorized as a detoxification response, leading to induction of enzymes involved in phase I/II metabolism and phase III exporters to promote elimination of triggering agents like LCA. An additional role of modulation of innate and adaptive immune responses by microbial bile salt metabolites via nuclear bile salt receptors is emerging [35]. Alternative nuclear bile salt receptors (with exemplary bile salt ligand in parentheses) include constitutive androstane receptor (CAR, 6-keto-LCA), liver X receptor (LXRα, pregnane X receptor (PXR, LCA), RAR-related orphan receptor gamma (RORγt, 3-keto LCA) and vitamin D receptor (VDR, 3-keto LCA) [33,35,36,37]. Recognized bile salts include oxidized (keto) forms of unconjugated LCA and stereoisomeric forms of unconjugated secondary bile salts. Activation of these receptors by conjugated bile salts is less well explored, and as such it remains to be determined if (oxidized) MBSCs can act as activating ligands of (some of) these receptors.

### 2.3. MBSCs Are Prone to Degradation by Host and Bacterial Enzymes

Summarizing the above section on bile salt receptors, experimental evidence supports that the MBSCs initially identified [9] can activate TGR5 and FXR, the latter likely requiring a transporter, e.g., NTCP, for efficient cellular entry of the compounds. At present, there is no conclusive evidence that MBSCs are absorbed in intact form. This outstanding question is all the more important given the known susceptibility of the *N*-amidate bond to enzymatic cleavage in bile salts conjugated with amino acids other than glycine or taurine. We tested whether MBSCs are deconjugated in vitro by pancreatic carboxypeptidases and bacterial BSH. Carboxypeptidases typically cleave the peptide bond at the carboxy terminal of peptides and protein but are also capable of hydrolyzing the amidate bond in conjugated bile salts. Pancreatic carboxypeptidase A is released in pancreatic juice, while pancreatic carboxypeptidase B circulates in the bloodstream and requires prior proteolytic activation. Both carboxypeptidases were found to hydrolyze the amidate bond in CA and CDCA conjugated with *L*-leucine, *L*-phenylalanine or *L*-tyrosine (Figure 3A,B). Under the experimental conditions, carboxypeptidase A completely degraded *L*-phenylalanine and *L*-tyrosine bile salt conjugates within 15–30 min, with carboxypeptidase B requiring longer reaction times to achieve this. Leucine-based MBSCs were deconjugated at a slower pace and showed partial resistance to degradation by carboxypeptidase B.

Bile salt hydrolase (BSH, a.k.a. choloylglycine hydrolase) is an intracellular enzyme widely expressed by bacteria inhabiting the gut. In line with expectations, host-derived GCA is not affected by the tested carboxypeptidases but is readily degraded by BSH in the slightly acidic pH range prevailing in the colonic environment (Figure 3C,D). Likewise, MBSCs containing an *L*-amino acid are sensitive to hydrolysis by BSH, with CDCA-based variants being cleaved more readily. Interestingly, MBSCs containing the *D*-enantiomer of tyrosine were fully resistant to cleavage by pancreatic carboxypeptidases and largely insensitive towards BSH-mediated hydrolysis. It is currently unknown if *D*-amino acids, which are produced by numerous gut bacteria, are substrates for MBSC formation. The initially described MBSCs showed chromatographic coelution with authentic standards (*L*-amino acid based), using a chromatographic setup that would likely have allowed separation of *L*- and *D*-amino acid variants of MBSCs. It is thus fair to assume that naturally occurring MBSCs contain at least *L*-amino acids. The above findings demonstrate that MBSCs are susceptible to cleavage by host and bacterial enzymes in vitro. Factors such as contact time, chyme/fecal matrix and pH likely influence deconjugation of MBSCs in situ, and the mere ability to detect MBSCs in feces demonstrates that enzymatic degradation is not complete. Degradation may, however, contribute to the apparent inability to detect MBSCs in the host circulation. Note that in the original study, no deconjugation of MBSCs was observed when added to an actively growing human fecal culture, whereas supplied GCA was readily degraded. The discrepancy with findings from our cell-free approach may be due to differential uptake/permeation of GCA and MBSCs and/or depend on the substrate specificity of the employed BSH.

### 2.4. MBSCs in Human Disease and Experimental Models

At present, insight into the occurrence of MBSCs in the context of disease stems from the initial report only and as such concerns LeuCA, PheCA and TyrCA [9]. These MBSCs were detected, i.e., matches were identified with spectra in public mass spectrometry datasets (GNPS database), in a number of studies comprising gastrointestinal samples from humans or mice. In fecal samples analyzed in the framework of the American Gut Project and representing a cross-section of a self-selected adult population, 1.6% of all fecal samples were found to have matches to one or multiple MBSCs [9]. In studies in infants and patients with inflammatory bowel disease (IBD) or cystic fibrosis, the proportion of fecal samples containing at least one MBSC was notably higher (ranging from circa 12% to 25%). In IBD, fecal abundance of all three MBSCs was higher in patients with Crohn’s disease relative to non-IBD controls, and the high abundance was associated with a dysbiotic state. A known MBSC producer (*Clostridium bolteae,* reclassified as *Enterocloster bolteae*) was identified earlier as more prevalent in the dysbiotic gut of patients with IBD [38]. Among patients with cystic fibrosis, the fraction of MBSC-positive fecal samples was notably higher in subjects with pancreatic insufficiency (circa 55%) than in those without (circa 25%). Pancreatic insufficiency results in a lack of pancreatic digestive enzymes and thus inability to properly digest food, with a large fraction of ingested nutrients entering the colon. Gut microbial changes in both these diseases likely impact gut luminal and fecal MBSC levels, with changes in both production and breakdown conceivable. In the context of pancreatic insufficiency, lack of pancreatic carboxypeptidase A may hamper luminal degradation of MBSCs by host enzymes.

In the context of experimental disease, information on MBSCs is also scarce thus far. Relative abundance of MBSCs was measured by LC-MS in mice fed a high-fat diet [9]. MBSCs were not present in feces of chow-fed mice but were readily detected after high-fat feeding, with LeuCA and PheCA being the dominant variants. Initiation of a high-fat diet resulted in large increases of basal levels of fecal MBSCs in atherosclerosis-prone mice from 2 weeks of feeding onwards. Here, increased abundance of *Clostridium* sp. following introduction of the high-fat diet was strongly associated with fecal levels of individual MBSCs. These observations add to the notion that alterations in gut microbial abundance and composition can affect fecal MBSC levels. It is currently unresolved if MBSC production in (experimental) disease conditions has a functional impact on the host and/or disease process or is merely a consequence of gut dysbiosis.

Using a quantitative approach, we made a preliminary assessment on the presence of MBSCs in intestinal content of patients with acute intestinal failure [39]. (Figure 4). LC-MS analysis of chyme retrieved from the jejunal stoma output of these patients revealed the presence of not only CA-based *N*-amidates but also CDCA-conjugates with Leu, Phe and Tyr (Lenicek, Koelfat, Schaap et al., unpublished observations). CA-based MBSCs appeared more abundant than CDCA-containing variants. Here, MBSCs comprised a minor fraction of total bile salts (8.7 [0.9–39.2] nmol/mmol). The simultaneous presence in vast larger amounts of other receptor activating bile salts, combined with FXR/TGR5-binding affinities that are more or less similar for MBSCs and non-MBSC species, raises the question if MBSCs have a relevant contribution to host bile salt signaling or if their function relates more to gut ecology.

### 2.5. Potential Roles of MBSCs in the Gut

Inherent to their detergent properties, bile salts have antibacterial activity, with physicochemical properties (e.g., hydrophobicity) an important determinant of bactericidal activity of individual species. Microbial metabolism of bile salts is widely viewed as a defensive mechanism that provides an advantage to adapted bacterial species. It is currently unresolved if microbial bile salt conjugation serves as an additional defensive strategy or has a more direct role in gut ecology. Common principles of microbial bile salt transformation are briefly discussed and viewed in light of MBSCs.

Bacterial deconjugation of bile salts results in more hydrophobic molecules, which are in general more toxic, especially when accompanied by 7α-dehydroxylation. At the same time, deconjugated bile salts have lower solubility in the fecal water phase, and precipitation, as well as enhanced binding to fibrous dietary matter [40], renders them harmless. In the colon, epithelial cells can passively take up deconjugated bile salts in their protonated form for circulation to the liver. This route contributes to the lowering of luminal bile salt levels by microbial deconjugation [41].

Microbial BSHs catalyze the deconjugation of bile salts. BSH activity is widespread in gut bacteria, especially in Gram-positive bacteria, that due to their lack of an outer membrane are more sensitive to detergent effects of bile salts [42]. Bacterial deconjugation of bile salts entering the colon is nearly quantitative, with only minor fractions of conjugated species found in feces. Capability for 7α-dehydroxylation, the hallmark reaction of secondary bile salt formation that must be preceded by deconjugation, is restricted to a few species of the obligate anaerobic order *Clostridiales* [43]. 7α-dehydroxylation is catalyzed by *bai* proteins and may play a role in niche restriction by forming hydrophobic secondary bile salts. This is exemplified by *Clostridium scindens*, which inhibits germination of the pathogen *Clostridium difficile* by producing secondary bile salts. Reconjugation may protect capable species from the growth-repressive effects of ‘free’ bile salts [44].

Generation of more hydrophilic species is a parallel strategy for microbial detoxification of bile salts. Reaction types include dehydrogenation/oxidation and epimerization of hydroxyl groups in the steroid nucleus. Site-specific enzymes that catalyze these conversions (e.g., hydroxysteroid dehydrogenases) are widespread in gut bacteria [8,45]. Oxidized or β-stereoisomers of bile salts have in general a lower affinity for FXR, which in the gut controls expression of antimicrobial peptides including cathelicidin [46,47]. Via this indirect route, microbial biotransformation of bile salts may increase bacterial survival in the intestines. Microbial conjugation of bile salts can be viewed as an additional tactic to increase hydrophilicity and render them less toxic for the producing strain. A further molecular adaption employed by, e.g., Gram-negative bacteria such as *Salmonella enterica* or *Escherichia coli* is expression of efflux pumps that maintain low levels of bile salts inside bacteria [48,49]. It is tempting to speculate that microbial conjugation results in better substrates for such export pumps.

Function of MBSCs as a niche factor was explored in the original publication [9]. Addition of MBSCs to an actively growing batch of human fecal bacteria had no effect on the community structure in vitro. Microbiome data has not been reported for mice gavaged with MBSCs, and it remains to be determined if MBSCs can act as a niche factor in vivo. To summarize, experimental insight is necessary to appreciate what bacteria gain from conjugation of bile acids, likely an energy-requiring reaction. Possibilities that can be considered include (i) lowering of intracellular levels of (toxic) bile acids by promoting their export to the surroundings, (ii) specific roles in bacterial metabolism, and (iii) generation of bile salt metabolites that are harmful to nonadapted bacteria to obtain a growth advantage, akin to [50].

## 3. Concluding Remarks

The pioneering study of Quinn et al. that was published in March 2020 resulted in the identification of three MBSCs and two *Clostridium bolteae* strains capable of synthesizing them. Initial explorations indicated that MBSCs were able to activate the bile salt receptor Fxr in the liver when administered to mice. Meanwhile, the number of bacterial strains capable of MBSC formation in vitro, as well as the diversity of MBSCs, has greatly expanded. Insight into the functions of MBSCs is, unsurprisingly, still lagging behind. Of particular interest would be to learn if MBSCs are absorbed by the host in intact form, in amounts that would make a relevant contribution to host signaling on top of abundant receptor-modulating bile salts in the circulation.

This is the more relevant, as MBSCs appear prone to degradation by bacterial and host enzymes that are also present in the gut lumen. Disease associations have been uncovered for fecal MBSCs and Crohn’s disease and cystic fibrosis and overnutrition-related disorders. Analysis of samples of the above patient groups may provide the first clues to a possible impact of MBSCs on the host and/or the disease process. If the consequence of an altered gut microbial community structure, MBSCs may have potential as a biomarker for gut dysbiosis. Conversely, MBSCs may engage in intermicrobial communication and have a function in gut ecology. Much is to be learned about MBSCs, and the future will tell if, like their equivalent in pop music, these new kids on the block are here to stay.

## Figures and Tables

**Figure 1 metabolites-12-00176-f001:**
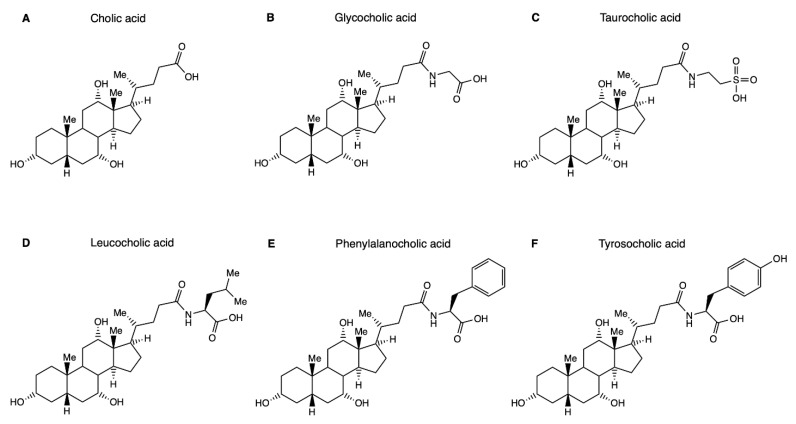
Structure of bile acids and host- and microbially conjugated variants. (**A**) Structure of the primary bile acid cholic acid that undergoes hepatic *N*-amidation with glycine or taurine to form glycocholic acid (**B**) or taurocholic acid (**C**). Specific gut microbial strains are capable of conjugating bile acids with a range of additional amino acids. Shown here are the three microbial bile salt conjugates that were discovered first, namely leucocholic acid (**D**), phenylalanocholic acid (**E**) and tyrosocholic acid (**F**). It is conceivable that the physicochemical properties of the amino acid employed for conjugation affect physicochemical and biological properties of the resultant bile salt conjugate.

**Figure 2 metabolites-12-00176-f002:**
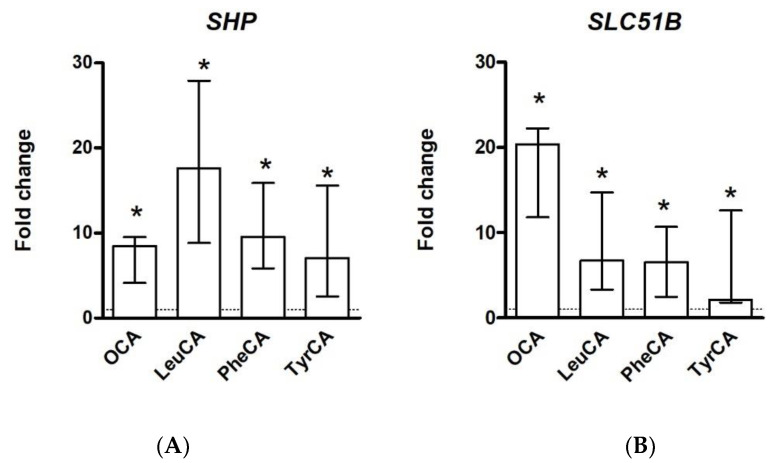
MBSCs induce FXR target gene expression in human HuH7 hepatoma cells. HuH7 cells overexpressing NTCP were grown until ca. 80% confluency. After overnight serum starvation, cells were treated for 6 h with solvent (0.1% DMSO) or the indicated test compounds (OCA: 10 µmol/L, MBSCs: 50 µmol/L). MBSCs were synthesized as described [9]. Gene expression was assessed by RTqPCR and is expressed relative to the control group (solvent-exposed cells, dotted line). MBSCs induce expression of FXR targets short heterodimer protein (SHP) (**A**) and solute carrier family 51B (SLC51B/OSTβ) (**B**). A representative experiment with 4 replicates per condition is depicted. Bars represent median and range (* denotes *p* < 0.05, Mann–Whitney U test).

**Figure 3 metabolites-12-00176-f003:**
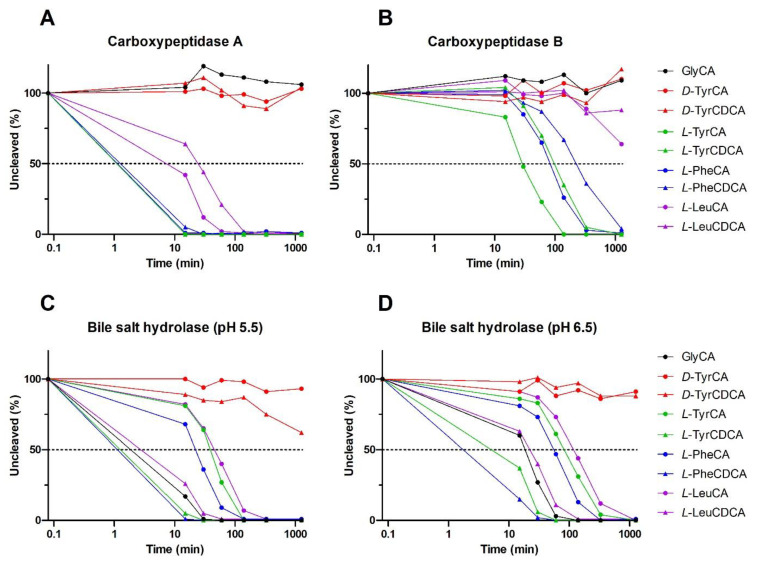
MBSCs are deconjugated by digestive and microbial enzymes in vitro. A mixture of the indicated bile salts (0.5 nmol each) was incubated at 37 °C with 5 units of either bovine pancreatic carboxypeptidase A (**A**), porcine pancreatic carboxypeptidase B (**B**) or bile salt hydrolase from *Clostridium perfringens* (**C**,**D**) in 0.5 mL of recommended buffers. Reactions were stopped at the indicated time points by transfer of a 50 µL aliquot to tubes containing 150 µL of HPLC-grade methanol, followed by further work-up for LC-MS analysis of remaining conjugate and liberated parent bile salt. Cleavage is expressed relative to the starting amount.

**Figure 4 metabolites-12-00176-f004:**
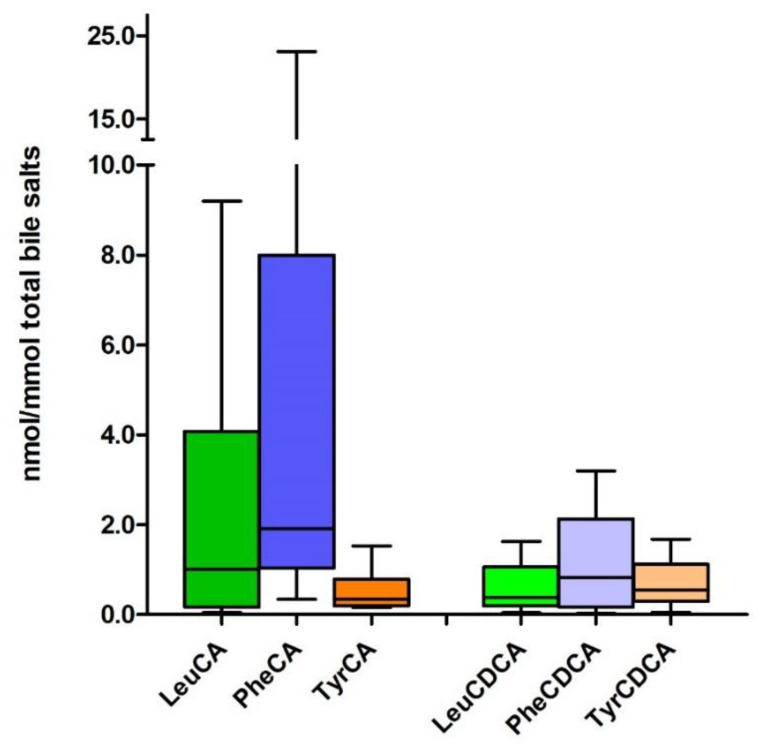
Known and heretofore unreported MBSCs are present in chyme of patients with acute intestinal failure. Chyme was collected from the stomal output of patients with intestinal failure and a temporary double enterostomy at the level of the jejunum/proximal ileum (*n* = 12). Samples were processed for LC-MS assay of bile salt composition. Authentic standards for the indicated MBSCs were synthesized and included as external standards in the LC-MS assay. Data are depicted as box and whisker plots, displaying median and minimum and maximum values.

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
