# Peer review of "New Kids on the Block: Bile Salt Conjugates of Microbial Origin"

_metabolites, 2022, doi:10.3390/metabo12020176_

Round 1
Reviewer 1 Report
The authors have submitted a manuscript to Metabolites journal that describes in a concise, clear, and elegant way the most important aspects related to MBCs, which are:
-Generalities on the origin of MBCs: from cholesterol towards natural, primary, secondary, conjugated bile acids and beyond.
- Enzymatic reactions developed by microbiota that can originate MBCs, as well as its degradation and enterohepatic recirculation.
- Cell carriers and efflux pumps localized in the enterohepatic circuit, which could be responsible of the uptake and efflux of MBCs.
-Membrane and nuclear MBCs receptors
- MBCs functions
-Metabolism, modification, and degradation pathways for MBCs
-MBCs role in the development and evolution of liver diseases.
I only have some curiosities and a minor comment to authors:
1. Authors indicate the possibility that MBCs may be substrates for NTCP. Although it is well known that NTCP is the main carrier for bile salts, I wonder if there is any bibliographic data or is there a possibility that MBCs could be taking up by proteins of the OATP family.
2. Regarding efflux pumps in the canalicular membrane, in addition to BSEP is there any data about MBCs could be exported into bile by MRP2 or BCRP?
3. Figure 2 (legend): I think it is necessary to indicate which is the control against which the different results are compared
I think that the article is of interest to be published in Metabolites journal since crosstalk between intestinal microbiota and enterohepatic tissues, through MBCs, could plays a very important role in the ethiopathogenesis of liver diseases and these species could be markers of liver or intestinal pathologies in the future, although some of the aspects must further investigated.
Reviewer 2 Report
The manuscript titled, ‘New kids on the block: bile salt conjugates of microbial origin’ is a timely review of novel bile acid transformations that have been recently discovered. Mostly, this review is based on 1 publication (Quinn et. al., 2020), and cites a few other papers that have discovered similar transformations by gut bacteria. Considering the paradigm shifting discovery by Quinn et. al., I am generally enthusiastic about this review and the valuable insight it provides to the reader about this new class of bile acid metabolites of bacterial origin. However, considering that this is a new field of bile acid biology, the literature depth required for this review is limited, and it generates more questions than answers them. I have some suggestions and questions for consideration by the authors that could improve the manuscript:
- In part 2.1., the authors suggest that bile acids do not permeate the small intestine. I assume the authors are specifically referring to N-amidated bile acids here, which should be clarified, because evidence suggests that passive diffusion of unconjugated bile acids can occur throughout the GI tract, including the small intestine. Further, in metabolic syndrome, gut permeability leads to paracellular diffusion of gut metabolites from the intestine into portal and systemic circulation. Recent studies measuring portal bile acids have found significant amounts of conjugated bile acids in portal circulation, which innervates the entire GI tract. Therefore, bile acids in the small intestine (conjugated and unconjugated) can and do permeate. Further clarification addressing these points would improve consistency with published literature.
- In 2.1. further, the authors suggest that a “high luminal concentration” of bile acids in “micellar phase” contributes to low number of bacteria in the small intestine. Again, it is assumed that they are referring to conjugated bile acids, which should be clarified. Especially because the cecum and colon contains higher concentration of bile acids than the small intestine. Therefore, high bile acid concentration leading to micelle formation would occur in the large intestine as well, where bacteria thrive. Low microbiota load in the small intestine is possibly driven by GI motility, pH, and oxygen concentrations rather than bile acid concentration and/or micelle formation. This section could use clarifications and corrections accordingly.
- An outstanding question that could be detailed in the review is why do bacteria conjugate bile acids? What do they gain from this? In section 2.2., the authors do address this question by suggesting that ‘ATP could be directly consumed’ in such a reaction. They also address this question briefly in section 3 and in the ‘Conclusion’. A separate section dedicated to this section will be valuable to the reader. Relatedly, are there other instances akin to this where gut bacteria add moieties to host metabolites for specific functions - perhaps to increase transport, or use in bacterial metabolism/growth, or to kill other commensals and gain a growth advantage? Including this information will likely generate new hypotheses addressing bile acid conjugation from a bacterial standpoint, separate from a host standpoint.
- In 3.1., the authors make statements about ASBT that need to be readdressed or detailed for clarification. Bile acids compete for ASBT binding, with certain amino acids having specific binding affinities for bile acids (PMID: 16608845). Further, studies have found differences in bile acid transport efficiencies for rat, mouse, and human ASBT (see PMID: 12456679 and PMID: 9458785 for example). A recent study found that monohydroxy secondary bile acid, LCA has a higher affinity for human ASBT than di- and tri-hydroxy bile acids (PMID: 33434516). Lastly, germ-free animals have high levels of ASBT, which is then reduced when colonized by gut bacteria, suggesting that bacterial metabolites could themselves alter ASBT expression levels. These aspects of ASBT need to be clarified, corrected, and discussed with more specificity and detail.
- Section 3.2. could benefit from a literature analysis of known bile acids that activate said receptors, and their EC50s that change when N-amidated. For example, see table in PMID: 18307294. Instances where bile acid affinities for receptors are discussed, following the statements with known EC50s (in the text or as a table) would substantiate their claims.
- In section 3.2.2., other nuclear hormone receptors that have shown to be important mediators of bile acid signaling (RORα, LXR) should also be discussed.
General points:
- The manuscript has data that substantiates some of the authors’ claims, which is appreciated. However, the manuscript has “data not shown” and “unpublished data” referenced on many occasions for vital pieces of information and conclusions. Considering that this is a review on recently published literature, any reference to unpublished data that has not been peer-reviewed should be avoided if possible.
- In many instances, the authors make blanketed statements, particularly with receptors and transporters that need to be readdressed (or specify whether the statement is true for mouse, or human, or a cell line). For example, in section 3.5., the authors claim that “colonic epithelial cells do not express ASBT”, which may be true in humans (?), but is not true for human colon cancer derived epithelial cell lines, in germ free mouse colon epithelium, and in conventional mouse and rat colons (where ASBT is expressed at low levels).
Reviewer 3 Report
The study is well done, the material is large enough and the methods look reliable. However the study is based on extensive and very recent literature, gives some new information and this warrants its publication.
Round 2
Reviewer 2 Report
The authors have done a great job with the revision. Please accept without any further changes.